# Improved Error Bounds for Tree Representations of Metric Spaces

**Samir Chowdhury**
Department of Mathematics
The Ohio State University
Columbus, OH 43210
`chowdhury.57@osu.edu`

**Facundo Mémoli**
Department of Mathematics
Department of Computer Science and Engineering
The Ohio State University
Columbus, OH 43210
`memoli@math.osu.edu`

**Zane Smith**
Department of Computer Science and Engineering
The Ohio State University
Columbus, OH 43210
`smith.9911@osu.edu`

## Abstract

Estimating optimal phylogenetic trees or hierarchical clustering trees from metric data is an important problem in evolutionary biology and data analysis. Intuitively, the goodness-of-fit of a metric space to a tree depends on its inherent treeness, as well as other metric properties such as intrinsic dimension. Existing algorithms for embedding metric spaces into tree metrics provide distortion bounds depending on cardinality. Because cardinality is a simple property of any set, we argue that such bounds do not fully capture the rich structure endowed by the metric. We consider an embedding of a metric space into a tree proposed by Gromov. By proving a stability result, we obtain an improved additive distortion bound depending only on the hyperbolicity and doubling dimension of the metric. We observe that Gromov's method is dual to the well-known single linkage hierarchical clustering (SLHC) method. By means of this duality, we are able to transport our results to the setting of SLHC, where such additive distortion bounds were previously unknown.

## 1 Introduction

Numerous problems in data analysis are formulated as the question of embedding high-dimensional metric spaces into "simpler" spaces, typically of lower dimension. In classical multidimensional scaling (MDS) techniques [18], the goal is to embed a space into two or three dimensional Euclidean space while preserving interpoint distances. Classical MDS is helpful in exploratory data analysis, because it allows one to find hidden groupings in amorphous data by simple visual inspection. Generalizations of MDS exist for which the target space can be a *tree metric space*—see [13] for a summary of some of these approaches, written from the point of view of *metric embeddings*. The metric embeddings literature, which grew out of MDS, typically highlights the algorithmic gains made possible by embedding a complicated metric space into a simpler one [13].

The special case of MDS where the target space is a tree has been of interest in phylogenetics for quite some time [19, 5]; the *numerical taxonomy problem* (NTP) is that of finding an *optimal* tree embedding for a given metric space $(X, d_X)$, i.e. a tree $(X, t_X)$ such that the *additive distortion*, defined as $\|d_X - t_X\|_{\ell^\infty(X \times X)}$, is minimal over all possible tree metrics on $X$. This problem turns out to be NP-hard [3]; however, a 3-approximation algorithm exists [3], and a variant of this problem,

that of finding an optimal *ultrametric* tree, can be solved in polynomial time [11]. An ultrametric tree is a rooted tree where every point is equidistant from the root—for example, ultrametric trees are the outputs of hierarchical clustering (HC) methods that show groupings in data across different resolutions. A known connection between HC and MDS is that the output ultrametric of *single linkage hierarchical clustering* (SLHC) is a 2-approximation to the optimal ultrametric tree embedding [16], thus providing a partial answer to the NTP. However, it appears that the existing line of work regarding NTP does not address the question of quantifying the $\ell^\infty$ distance between a metric $(X, d_X)$ and its optimal tree metric, or even the optimal ultrametric. More specifically, we can ask:

**Question 1.** *Given a set $X$, a metric $d_X$, and an optimal tree metric $t_X^{\mathrm{opt}}$ (or an optimal ultrametric $u_X^{\mathrm{opt}}$), can one find a nontrivial upper bound on $\|d_X - t_X^{\mathrm{opt}}\|_{\ell^\infty(X \times X)}$ (resp. $\|d_X - u_X^{\mathrm{opt}}\|_{\ell^\infty(X \times X)}$) depending on properties of the metric $d_X$?*

The question of distortion bounds is treated from a different perspective in the discrete algorithms literature. In this domain, tree embeddings are typically described with *multiplicative* distortion bounds (described in §2) depending on the cardinality of the underlying metric space, along with (typically) pathological counterexamples showing that these bounds are tight [4, 10]. We remark immediately that (1) multiplicative distortion is distinct from the additive distortion encountered in the NTP, and (2) these embeddings are rarely used in machine learning, where HC and MDS methods are the main workhorses. Moreover, such multiplicative distortion bounds do not take two considerations into account: (1) the ubiquitousness of very large data sets means that a bound dependent on cardinality is not desirable, and (2) "nice" properties such as low intrinsic dimensionality or treeness of real-world datasets are not exploited in cardinality bounds.

We prove novel additive distortion bounds for two methods of tree embeddings: one into general trees, and one into ultrametric trees. These additive distortion bounds take into account (1) whether the data is treelike, and (2) whether the data has low doubling dimension, which is a measure of its intrinsic dimension. Thus we prove an answer to Question 1 above, namely, that the approximation error made by an optimal tree metric (or optimal ultrametric) can be bounded nontrivially.

**Remark 1.** The trivial upper bound is $\|d_X - t_X^{\mathrm{opt}}\|_{\ell^\infty(X \times X)} \leq \mathrm{diam}(X, d_X)$. To see this, observe that any ultrametric is a tree, and that SLHC yields an ultrametric $u_X$ that is bounded above by $d_X$.

**An overview of our approach.** A common measure of treeness is Gromov's $\delta$-hyperbolicity, which is a local condition on 4-point subsets of a metric space. Hyperbolicity has been shown to be a useful statistic for evaluating the quality of trees in [7]. The starting point for our work is a method used by Gromov to embed metric spaces into trees, which we call *Gromov's embedding* [12]. A known result, which we call *Gromov's embedding theorem*, is that if every 4-point subset of an $n$-point metric space is $\delta$-hyperbolic, then the metric space embeds into a tree with $\ell^\infty$ distortion bounded above by $2\delta \log_2(2n)$. The proof proceeds by a *linkage argument*, i.e. by invoking the definition of hyperbolicity at different scales along chains of points. By virtue of the embedding theorem, one can argue that hyperbolicity is a measure of the "treeness" of a given metric space. It has been shown in [1, 2] that various real-world data sets, such as Internet latencies and biological, social, and collaboration networks are inherently treelike, i.e. have low hyperbolicity. Thus, by Gromov's result, these real-world data sets can be embedded into trees with additive distortion controlled by their respective cardinalities. The cardinality bound might of course be undesirable, especially for very large data sets such as the Internet. However, it has been claimed without proof in [1] that Gromov's embedding can yield a 3-approximation to the NTP, independent of [3].

We note that the assumption of a metric input is not apparent in Gromov's embedding theorem. Moreover, the proof of the theorem does not utilize any metric property. This leads one to hope for bounds where the dependence on cardinality is replaced by a dependence on some *metric* notion. A natural candidate for such a metric notion is the *doubling dimension* of a space [15], which has already found applications in learning [17] and algorithm design [15]. In this paper, we present novel upper bounds on the additive distortion of a Gromov embedding, depending only on the hyperbolicity and doubling dimension of the metric space.

Our main tool is a stability theorem that we prove using a *metric induced by a Voronoi partition*. This result is then combined with the results of Gromov's linkage argument. Both the stability theorem and Gromov's theorem rely on the embedding satisfying a particular *linkage condition*, which can be described as follows: for any embedding $f : (X, d_X) \to (X, t_X)$, and any $x, x' \in X$, we have $t_X(x, x') = \max_c \min_i \Psi(x_i, x_{i+1})$, where $c = \{x_i\}_{i=0}^k$ is a chain of points joining $x$ to $x'$ and $\Psi$

is some function of $d_X$. A dual notion is to flip the order of the $\max, \min$ operations. Interestingly, under the correct objective function $\Psi$, this leads to the well-studied notion of SLHC. By virtue of this duality, the arguments of both the stability theorem and the scaling theorem apply in the SLHC setting. We introduce a new metric space statistic that we call *ultrametricity* (analogous to hyperbolicity), and are then able to obtain novel lower bounds, depending only on doubling dimension and ultrametricity, for the distortion incurred by a metric space when embedding into an ultrametric tree via SLHC.

We remark that just by virtue of the duality between Gromov's embedding and the SLHC embedding, it is possible to obtain a distortion bound for SLHC depending on cardinality. We were unable to find such a bound in the existing HC literature, so it appears that even the knowledge of this duality, which bridges the domains of HC and MDS methods, is not prevalent in the community.

The paper is organized as follows. The main thrust of our work is explained in §1. In §2 we develop the context of our work by highlighting some of the surrounding literature. We provide all definitions and notation, including the Voronoi partition construction, in §3. In §4 we describe Gromov's embedding and present Gromov's distortion bound in Theorem 3. Our contributions begin with Theorem 4 in §4 and include all the results that follow: namely the stability results in §5, the improved distortion bounds in §6, and the proof of tightness in §7.

The supplementary material contains (1) an appendix with proofs omitted from the body, (2) a practical demonstration in §A where we apply Gromov's embedding to a bitmap image of a tree and show that our upper bounds perform better than the bounds suggested by Gromov's embedding theorem, and (3) Matlab `.m` files containing demos of Gromov's embedding being applied to various images of trees.

## 2 Related Literature

MDS is explained thoroughly in [18]. In metric MDS [18] one attempts to find an embedding of the data $X$ into a low dimensional Euclidean space given by a point cloud $Y \subset \mathbb{R}^d$ (where often $d = 2$ or $d = 3$) such that the metric distortion (measured by the Frobenius norm of the difference of the Gram matrices of $X$ and $Y$) is minimized. The most common non-metric variant of MDS is referred to as *ordinal embedding*, and has been studied in [14].

A common problem with metric MDS is that when the intrinsic dimension of the data is higher than the embedding dimension, the clustering in the original data may not be preserved [21]. One variant of MDS that preserves clusters is the tree preserving embedding [20], where the goal is to preserve the single linkage (SL) dendrogram from the original data. This is especially important for certain types of biological data, for the following reasons: (1) due to speciation, many biological datasets are inherently "treelike", and (2) the SL dendrogram is a 2-approximation to the optimal ultrametric tree embedding [16], so intuitively, preserving the SL dendrogram preserves the "treeness" of the data. Preserving the treeness of a metric space is related to the notion of finding an optimal embedding into a tree, which ties back to the numerical taxonomy problem. The SL dendrogram is an embedding of a metric space into an ultrametric tree, and can be used to find the optimal ultrametric tree [8].

The quality of an embedding is measured by computing its *distortion*, which has different definitions in different domain areas. Typically, a *tree embedding* is defined to be an injective map $f : X \to Y$ between metric spaces $(X, d_X)$ and $(Y, t_Y)$, where the target space is a tree. We have defined the additive distortion of a tree embedding in an $\ell^\infty$ setting above, but $\ell^p$ notions, for $p \in [1, \infty)$, can also be defined. Past efforts to embed a metric into a tree with low additive distortion are described in [19, Chapter 7]. One can also define a *multiplicative distortion* [4, 10], but this is studied in the domain of discrete algorithms and is not our focus in the current work.

## 3 Preliminaries on metric spaces, distances, and doubling dimension

A finite metric space $(X, d_X)$ is a finite set $X$ together with a function $d_X : X \times X \to \mathbb{R}_+$ such that: (1) $d_X(x, x') = 0 \iff x = x'$, (2) $d_X(x, x') = d_X(x', x)$, and (3) $d_X(x, x') \leq d_X(x, x'') + d_X(x'', x')$ for any $x, x', x'' \in X$. A pointed metric space is a triple $(X, d_X, p)$, where $(X, d_X)$ is a finite metric space and $p \in X$. All the spaces we consider are assumed to be finite.

For a metric space $(X, d_X)$, the *diameter* is defined to be $\mathrm{diam}(X, d_X) := \max_{x,x' \in X} d_X(x, x')$. The *hyperbolicity* of $(X, d_X)$ was defined by Gromov [12] as follows:

$$\mathrm{hyp}(X, d_X) := \max_{x_1, x_2, x_3, x_4 \in X} \Psi_X^{\mathrm{hyp}}(x_1, x_2, x_3, x_4), \text{ where}$$

$$\Psi_X^{\mathrm{hyp}}(x_1, x_2, x_3, x_4) := \tfrac{1}{2}\Big( d_X(x_1, x_2) + d_X(x_3, x_4)$$
$$- \max\big( d_X(x_1, x_3) + d_X(x_2, x_4), d_X(x_1, x_4) + d_X(x_2, x_3)\big)\Big).$$

A tree metric space $(X, t_X)$ is a finite metric space such that $\mathrm{hyp}(X, t_X) = 0$ [19]. In our work, we strengthen the preceding characterization of trees to the special class of ultrametric trees. Recall that an ultrametric space $(X, u_X)$ is a metric space satisfying the *strong triangle inequality*:

$$u_X(x, x') \leq \max(u_X(x, x''), u_X(x'', x')), \forall x, x', x'' \in X.$$

**Definition 1.** We define the *ultrametricity* of a metric space $(X, d_X)$ as:

$$\mathrm{ult}(X, d_X) := \max_{x_1, x_2, x_3 \in X} \Psi_X^{\mathrm{ult}}(x_1, x_2, x_3), \text{ where}$$

$$\Psi_X^{\mathrm{ult}}(x_1, x_2, x_3) := d_X(x_1, x_3) - \max\big( d_X(x_1, x_2), d_X(x_2, x_3)\big).$$

We introduce ultrametricity to quantify the deviation of a metric space from being ultrametric. Notice that $(X, u_X)$ is an ultrametric space if and only if $\mathrm{ult}(X, u_X) = 0$. One can verify that an ultrametric space is a tree metric space.

We will denote the cardinality of a set $X$ by writing $|X|$. Given a set $X$ and two metrics $d_X, d_X'$ defined on $X \times X$, we denote the $\ell^\infty$ distance between $d_X$ and $d_X'$ as follows:

$$\|d_X - d_X'\|_{\ell^\infty(X \times X)} := \max_{x, x' \in X} |d_X(x, x') - d_X'(x, x')|.$$

We use the shorthand $\|d_X - d_X'\|_\infty$ to mean $\|d_X - d_X'\|_{\ell^\infty(X \times X)}$. We write $\approx$ to mean "approximately equal to." Given two functions $f, g : \mathbb{N} \to \mathbb{R}$, we will write $f \asymp g$ to mean asymptotic tightness, i.e. that there exist constants $c_1, c_2$ such that $c_1|f(n)| \leq |g(n)| \leq c_2|f(n)|$ for sufficiently large $n \in \mathbb{N}$.

**Induced metrics from Voronoi partitions.** A key ingredient of our stability result involves a Voronoi partition construction. Given a metric space $(X, d_X)$ and a subset $A \subseteq X$, possibly with its own metric $d_A$, we can define a new metric $d_X^A$ on $X \times X$ using a Voronoi partition. First write $A = \{x_1, \ldots, x_n\}$. For each $1 \leq i \leq n$, we define $\widetilde{V}_i := \{x \in X : d_X(x, x_i) \leq \min_{j \neq i} d_X(x, x_j)\}$. Then $X = \bigcup_{i=1}^n \widetilde{V}_i$. Next we perform the following disjointification trick:

$$V_1 := \widetilde{V}_1, \ V_2 := \widetilde{V}_2 \setminus \widetilde{V}_1, \ldots, V_n := \widetilde{V}_n \setminus \Big( \bigcup_{i=1}^{n-1} \widetilde{V}_i \Big).$$

Then $X = \bigsqcup_{i=1}^n V_i$, a disjoint union of *Voronoi cells* $V_i$.

Next define the *nearest-neighbor map* $\eta : X \to A$ by $\eta(x) = x_i$ for each $x \in V_i$. The map $\eta$ simply sends each $x \in X$ to its closest neighbor in $A$, up to a choice when there are multiple nearest neighbors. Then we can define a new (pseudo)metric $d_X^A : X \times X \to \mathbb{R}_+$ as follows:

$$d_X^A(x, x') := d_A(\eta(x), \eta(x')).$$

Observe that $d_X^A(x, x') = 0$ if and only if $x, x' \in V_i$ for some $1 \leq i \leq n$. Symmetry also holds, as does the triangle inequality.

A special case of this construction occurs when $A$ is an $\varepsilon$-net of $X$ endowed with a restriction of the metric $d_X$. Given a finite metric space $(X, d_X)$, an *$\varepsilon$-net* is a subset $X^\varepsilon \subset X$ such that: (1) for any $x \in X$, there exists $s \in X^\varepsilon$ such that $d_X(x, s) < \varepsilon$, and (2) for any $s, s' \in X^\varepsilon$, we have $d_X(s, s') \geq \varepsilon$ [15]. For notational convenience, we write $d_X^\varepsilon$ to refer to $d_X^{X^\varepsilon}$. In this case, we obtain:

$$\|d_X - d_X^\varepsilon\|_{\ell^\infty(X \times X)} = \max_{x, x' \in X} \left| d_X(x, x') - d_X^\varepsilon(x, x') \right|$$
$$= \max_{1 \leq i, j \leq n} \max_{x \in V_i, x' \in V_j} \left| d_X(x, x') - d_X^\varepsilon(x, x') \right|$$
$$= \max_{1 \leq i, j \leq n} \max_{x \in V_i, x' \in V_j} \left| d_X(x, x') - d_X(x_i, x_j) \right|$$
$$\leq \max_{1 \leq i, j \leq n} \max_{x \in V_i, x' \in V_j} \big( d_X(x, x_i) + d_X(x', x_j)\big) \leq 2\varepsilon. \qquad (1)$$

**Covering numbers and doubling dimension.** For a finite metric space $(X, d_X)$, the open ball of radius $\varepsilon$ centered at $x \in X$ is denoted $B(x, \varepsilon)$. The $\varepsilon$-*covering number* of $(X, d_X)$ is defined as:

$$N_X(\varepsilon) := \min\left\{n \in \mathbb{N} : X \subset \bigcup_{i=1}^{n} B(x_i, \varepsilon) \text{ for } x_1, \ldots, x_n \in X\right\}.$$

Notice that the $\varepsilon$-covering number of $X$ is always bounded above by the cardinality of an $\varepsilon$-net. A related quantity is the *doubling dimension* $\mathrm{ddim}(X, d_X)$ of a metric space $(X, d_X)$, which is defined to be the minimal value $\rho$ such that any $\varepsilon$-ball in $X$ can be covered by at most $2^\rho$ $\varepsilon/2$-balls [15]. The covering number and doubling dimension of a metric space $(X, d_X)$ are related as follows:

**Lemma 2.** *Let $(X, d_X)$ be a finite metric space with doubling dimension bounded above by $\rho > 0$. Then for all $\varepsilon \in (0, \mathrm{diam}(X)]$, we have $N_X(\varepsilon) \leq \big(8\,\mathrm{diam}(X)/\varepsilon\big)^\rho$.*

## 4 Duality between Gromov's embedding and SLHC

Given a metric space $(X, d_X)$ and any two points $x, x' \in X$, we define a *chain* from $x$ to $x'$ over $X$ as an ordered set of points in $X$ starting at $x$ and ending at $x'$:

$$c = \{x_0, x_1, x_2, \ldots, x_n : x_0 = x, x_n = x', x_i \in X \text{ for all } 0 \leq i \leq n\}.$$

The set of all chains from $x$ to $x'$ over $X$ will be denoted $C_X(x, x')$. The *cost* of a chain $c = \{x_0 \ldots, x_n\}$ over $X$ is defined to be $\mathrm{cost}_X(c) := \max_{0 \leq i < n} d_X(x_i, x_{i+1})$.

For any metric space $(X, d_X)$ and any $p \in X$, the *Gromov product of $X$ with respect to $p$* is a map $g_{X,p} : X \times X \to \mathbb{R}_+$ defined by:

$$g_{X,p}(x, x') := \tfrac{1}{2}\big(d_X(x, p) + d_X(x', p) - d_X(x, x')\big).$$

We can define a map $g_{X,p}^{\mathcal{T}} : X \times X \to \mathbb{R}_+$ as follows:

$$g_{X,p}^{\mathcal{T}}(x, x')_p := \max_{c \in C_X(x,x')} \min_{x_i, x_{i+1} \in c} g_{X,p}(x_i, x_{i+1}).$$

This induces a new metric $t_{X,p} : X \times X \to \mathbb{R}_+$:

$$t_{X,p}(x, x') := d_X(x, p) + d_X(x', p) - 2g_{X,p}^{\mathcal{T}}(x, x').$$

Gromov observed that the space $(X, t_{X,p})$ is a tree metric space, and that $t_{X,p}(x, x') \leq d_X(x, x')$ for any $x, x' \in X$ [12]. This yields the trivial upper bound:

$$\|d_X - t_X\|_\infty \leq \mathrm{diam}(X, d_X).$$

The Gromov embedding $\mathcal{T}$ is defined for any pointed metric space $(X, d_X, p)$ as $\mathcal{T}(X, d_X, p) := (X, t_{X,p})$. Note that each choice of $p \in X$ will yield a tree metric $t_{X,p}$ that depends, *a priori*, on $p$.

**Theorem 3** (Gromov's embedding theorem [12]). *Let $(X, d_X, p)$ be an $n$-point pointed metric space, and let $(X, t_{X,p}) = \mathcal{T}(X, d_X, p)$. Then,*

$$\|t_{X,p} - d_X\|_{l^\infty(X \times X)} \leq 2\log_2(2n)\,\mathrm{hyp}(X, d_X).$$

Gromov's embedding is an MDS method where the target is a tree. We observe that its construction is dual—in the sense of swapping $\max$ and $\min$ operations—to the construction of the ultrametric space produced as an output of SLHC. Recall that the SLHC method $\mathcal{H}$ is defined for any metric space $(X, d_X)$ as $\mathcal{H}(X, d_X) = (X, u_X)$, where $u_X : X \times X \to \mathbb{R}_+$ is the ultrametric defined below:

$$u_X(x, x') := \min_{c \in C_X(x,x')} \mathrm{cost}_X(c).$$

As a consequence of this duality, we can bound the additive distortion of SLHC as below:

**Theorem 4.** *Let $(X, d_X)$ be an $n$-point metric space, and let $(X, u_X) = \mathcal{H}(X, d_X)$. Then we have:*

$$\|d_X - u_X\|_{\ell^\infty(X \times X)} \leq \log_2(2n)\,\mathrm{ult}(X, d_X).$$

*Moreover, this bound is asymptotically tight.*

The proof of Theorem 4 proceeds by invoking the definition of ultrametricity at various scales along chains of points; we provide details in Appendix B. We remark that the bounds in Theorems 3, 4 depend on both a local (ultrametricity/hyperbolicity) and a global property (cardinality); however, a natural improvement would be to exploit a global property that takes into account the metric structure of the underlying space. The first step in this improvement is to prove a set of *stability theorems*.

# 5 Stability of SLHC and Gromov's embedding

It is known that SLHC is robust to small perturbations of the input data with respect to the Gromov-Hausdorff distance between metric spaces, whereas other HC methods, such as average linkage and complete linkage, do not enjoy this stability [6]. We prove a particular stability result for SLHC involving the $\ell^\infty$ distance, and then we exploit the duality observed in §4 to prove a similar stability result for Gromov's embedding.

**Theorem 5.** *Let $(X, d_X)$ be a metric space, and let $(A, d_A)$ be any subspace with the restriction metric $d_A := d_X|_{A \times A}$. Let $\mathcal{H}$ denote the SLHC method. Write $(X, u_X) = \mathcal{H}(X, d_X)$ and $(A, u_A) = \mathcal{H}(A, d_A)$. Also write $u_X^A(x, x') := u_A(\eta(x), \eta(x'))$ for $x, x' \in X$. Then we have:*

$$\|\mathcal{H}(X, d_X) - \mathcal{H}(A, d_A)\|_\infty := \|u_X - u_X^A\|_\infty \leq \|d_X - d_X^A\|_\infty.$$

**Theorem 6.** *Let $(X, d_X, p)$ be a pointed metric space, and let $(A, d_A, a)$ be any subspace with the restriction metric $d_A := d_X|_{A \times A}$ such that $\eta(p) = a$. Let $\mathcal{T}$ denote the Gromov embedding. Write $(X, t_{X,p}) = \mathcal{T}(X, d_X, p)$ and $(A, t_{A,a}) = \mathcal{T}(A, d_A, a)$. Also write $t_{X,p}^A(x, x') := t_{A,a}(\eta(x), \eta(x'))$ for $x, x' \in X$. Then we have:*

$$\|\mathcal{T}(X, d_X, p) - \mathcal{T}(A, d_A, a)\|_\infty := \|t_{X,p} - t_{X,p}^A\|_\infty \leq 5\|d_X - d_X^A\|_\infty.$$

The proofs for both of these results use similar techniques, and we present them in Appendix B.

# 6 Improvement via Doubling Dimension

Our main theorems, providing additive distortion bounds for Gromov's embedding and for SLHC, are stated below. The proofs for both theorems are similar, so we only present that of the former.

**Theorem 7.** *Let $(X, d_X)$ be a $n$-point metric space with doubling dimension $\rho$, hyperbolicity $\mathrm{hyp}(X, d_X) = \delta$, and diameter $D$. Let $p \in X$, and write $(X, t_X) = \mathcal{T}(X, d_X, p)$. Then we obtain:*

$$\textit{Covering number bound:} \qquad \|d_X - t_X\|_\infty \leq \min_{\varepsilon \in (0, D]} \left( 12\varepsilon + 2\delta \log_2(2N_X(\varepsilon)) \right). \qquad (2)$$

*Also suppose $D \geq \frac{\delta\rho}{6 \ln 2}$. Then,*

$$\textit{Doubling dimension bound:} \qquad \|d_X - t_X\|_\infty \leq 2\delta + 2\delta\rho \left( \frac{13}{2} + \log_2 \left( \frac{D}{\delta\rho} \right) \right). \qquad (3)$$

**Theorem 8.** *Let $(X, d_X)$ be a $n$-point metric space with doubling dimension $\rho$, ultrametricity $\mathrm{ult}(X, d_X) = \nu$, and diameter $D$. Write $(X, u_X) = \mathcal{H}(X, d_X)$. Then we obtain:*

$$\textit{Covering number bound:} \qquad \|d_X - u_X\|_\infty \leq \min_{\varepsilon \in (0, D]} \left( 4\varepsilon + \nu \log_2(2N_X(\varepsilon)) \right). \qquad (4)$$

*Also suppose $D \geq \frac{\nu\rho}{4 \ln 2}$. Then,*

$$\textit{Doubling dimension bound:} \qquad \|d_X - u_X\|_\infty \leq \nu + \nu\rho \left( 6 + \log_2 \left( \frac{D}{\nu\rho} \right) \right). \qquad (5)$$

**Remark 9** (A remark on the NTP)**.** We are now able to return to Question 1 and provide some answers. Consider a metric space $(X, d_X)$. We can upper bound $\|d_X - u_X^{\mathrm{opt}}\|_\infty$ using the bounds in Theorem 8, and $\|d_X - t_X^{\mathrm{opt}}\|_\infty$ using the bounds in Theorem 7.

**Remark 10** (A remark on parameters)**.** Notice that as hyperbolicity $\delta$ approaches 0 (or ultrametricity approaches 0), the doubling dimension bounds (hence the covering number bounds) approach 0. Also note that as $\varepsilon \downarrow 0$, we get $N_X(\varepsilon) \uparrow |X|$, so Theorems 7,8 reduce to Theorems 3,4. Experiments lead us to believe that the interesting range of $\varepsilon$ values is typically a subinterval of $(0, D]$.

*Proof of Theorem 7.* Fix $\varepsilon \in (0, D]$ and let $X^\varepsilon = \{x_1, x_2, ..., x_k\}$ be a collection of $k = N_X(\varepsilon)$ points that form an $\varepsilon$-net of $X$. Then we may define $d_X^\varepsilon$ and $t_X^\varepsilon$ on $X \times X$ as in §3. Subsequent application of Theorem 3 and Lemma 2 gives the bound

$$\|d_X^\varepsilon - t_X^\varepsilon\|_{\ell^\infty(X \times X)} \leq \|d_{X^\varepsilon} - t_{X^\varepsilon}\|_{\ell^\infty(X^\varepsilon \times X^\varepsilon)} \leq 2\delta \log_2(2k) \leq 2\delta \log_2(2C\varepsilon^{-\rho}),$$

where we define $C := (8D)^\rho$. Then by the triangle inequality for the $\ell^\infty$ distance, the stability of $\mathcal{T}$ (Theorem 6), and using the result that $\|d_X - d_X^\varepsilon\|_{\ell^\infty(X \times X)} \leq 2\varepsilon$ (Inequality 1), we get:

$$\begin{aligned}
\|d_X - t_X\|_\infty &\leq \|d_X - d_X^\varepsilon\|_\infty + \|d_X^\varepsilon - t_X^\varepsilon\|_\infty + \|t_X^\varepsilon - t_X\|_\infty \\
&\leq 6\|d_X - d_X^\varepsilon\|_\infty + \|d_X^\varepsilon - t_X^\varepsilon\|_\infty \\
&\leq 12\varepsilon + 2\delta \log_2(2N_X(\varepsilon)).
\end{aligned}$$

Since $\varepsilon \in (0, D]$ was arbitrary, this suffices to prove Inequality 2. Applying Lemma 2 yields:

$$\|d_X - t_X\|_\infty \leq 12\varepsilon + 2\delta \log_2(2C\varepsilon^{-\rho}).$$

Notice that $C\varepsilon^{-\rho} \geq N_X(\varepsilon) \geq 1$, so the term on the right of the inequality above is positive. Consider the function

$$f(\varepsilon) = 12\varepsilon + 2\delta + 2\delta \log_2 C - 2\delta\rho \log_2 \varepsilon.$$

The minimizer of this function is obtained by taking a derivative with respect to $\varepsilon$:

$$f'(\varepsilon) = 12 - \frac{2\delta\rho}{\varepsilon \ln 2} = 0 \implies \varepsilon = \frac{\delta\rho}{6 \ln 2}.$$

Since $\varepsilon$ takes values in $(0, D]$, and $\lim_{\varepsilon \to 0} f(\varepsilon) = +\infty$, the value of $f(\varepsilon)$ is minimized at $\min(D, \frac{\delta\rho}{6\ln 2})$. By assumption, $D \geq \frac{\delta\rho}{6\ln 2}$. Since $\|d_X - t_X\|_\infty \leq f(\varepsilon)$ for all $\varepsilon \in (0, D]$, it follows that:

$$\|d_X - t_X\|_\infty \leq f\left(\frac{\delta\rho}{6\ln 2}\right) = \frac{2\delta\rho}{\ln 2} + 2\delta + 2\delta\rho \log_2\left(\frac{48D\ln 2}{\delta\rho}\right) \leq 2\delta + 2\delta\rho\left(\frac{13}{2} + \log_2\left(\frac{D}{\delta\rho}\right)\right). \qquad \square$$

## 7    Tightness of our bounds in Theorems 7 and 8

By the construction provided below, we show that our covering number bound for the distortion of SLHC is asymptotically tight. A similar construction can be used to show that our covering number bound for Gromov's embedding is also asymptotically tight.

**Proposition 11.** *There exists a sequence $(X_n, d_{X_n})_{n \in \mathbb{N}}$ of finite metric spaces such that as $n \to \infty$,*

$$\|d_{X_n} - u_{X_n}\|_\infty \asymp \min_{\varepsilon \in (0, D_n]} \left(4\varepsilon + \nu_n \log_2(2N_{X_n}(\varepsilon))\right) \to 0.$$

*Here we have written $(X_n, u_{X_n}) = \mathcal{H}(X_n, d_{X_n})$, $\nu_n = \mathrm{ult}(X_n, d_{X_n})$, and $D_n = \mathrm{diam}(X_n, d_{X_n})$.*

*Proof of Proposition 11.* After defining $X_n$ for $n \in \mathbb{N}$ below, we will denote the error term, our covering number upper bound, and our Gromov-style upper bound as follows:

$$E_n := \|d_{X_n} - u_{X_n}\|_\infty, \quad B_n := \min_{\varepsilon \in (0, D_n]} \rho(n, \varepsilon), \quad G_n := \log_2(2|X_n|)\, \mathrm{ult}(X_n, d_{X_n}), \text{ where}$$

$\rho : \mathbb{N} \times [0, \infty) \to \mathbb{R}$ is defined by $\rho(n, \varepsilon) = 4\varepsilon + \nu_n \log_2(2N_{X_n}(\varepsilon))$.

Here we write $|S|$ to denote the cardinality of a set $S$. Recall that the *separation* of a finite metric space $(X, d_X)$ is the quantity $\mathrm{sep}(X, d_X) := \min_{x \neq x' \in X} d_X(x, x')$. Let $(V, u_V)$ be the finite ultrametric space consisting of two equidistant points with common distance 1. For each $n \in \mathbb{N}$, let $L_n$ denote the *line metric space* obtained by choosing $(n+1)$ equally spaced points with separation $\frac{1}{n^2}$ from the interval $[0, \frac{1}{n}]$, and endowing this set with the restriction of the Euclidean metric, denoted $d_{L_n}$. One can verify that $\mathrm{ult}(L_n, d_{L_n}) \approx \frac{1}{2n}$. Finally, for each $n \in \mathbb{N}$ we define $X_n := V \times L_n$, and endow $X_n$ with the following metric:

$$d_{X_n}\big((v, l), (v', l')\big) := \max\big(d_V(v, v'), d_{L_n}(l, l')\big), \quad v, v' \in V,\ l, l' \in L_n.$$

**Claim 1.** $\mathrm{ult}(X_n, d_{X_n}) = \mathrm{ult}(L_n, d_{L_n}) \approx \frac{1}{2n}$. For a proof, see Appendix B.

**Claim 2.** $E_n \asymp \mathrm{diam}(L_n, d_{L_n}) = \frac{1}{n}$. To see this, let $n \in \mathbb{N}$, and let $x = (v, l), x' = (v', l') \in X_n$ be two points realizing $E_n$. Suppose first that $v = v'$. Then an optimal chain from $(v, l), (v, l')$ only

has to incur the cost of moving along the $L_n$ coordinate. As such, we obtain $u_{X_n}(x, x') \leq \frac{1}{n^2}$, with equality if and only if $l \neq l'$. Then,

$$E_n = \max_{x,x' \in X_n} |d_{X_n}(x, x') - u_{X_n}(x, x')| = \max_{l,l' \in L_n} |d_{L_n}(l, l') - \frac{1}{n^2}| = \frac{1}{n} - \frac{1}{n^2} \asymp \frac{1}{n}.$$

Note that the case $v \neq v'$ is not allowed, because then we would obtain $d_{X_n}(x, x') = d_V(v, v') = u_{X_n}(x, x')$, since $\text{sep}(V, d_V) \geq \text{diam}(L_n, d_{L_n})$ and all the points in $V$ are equidistant. Thus we would obtain $|d_{X_n}(x, x') - u_{X_n}(x, x')| = 0$, which is a contradiction because we assumed that $x, x'$ realize $E_n$.

**Claim 3.** For each $n \in \mathbb{N}, \varepsilon \in (0, D_n]$, we have:

$$N_{X_n}(\varepsilon) = \begin{cases} N_V(\varepsilon) & : \varepsilon > \text{sep}(V, d_V), \\ |V| & : \text{diam}(L_n, d_{L_n}) < \varepsilon \leq \text{sep}(V, d_V), \\ |V|N_{L_n}(\varepsilon) & : \varepsilon \leq \text{diam}(L_n, d_{L_n}). \end{cases}$$

To see this, note that in the first two cases, any $\varepsilon$-ball centered at a point $(v, l)$ automatically contains all of $\{v\} \times L_n$, so $N_{X_n}(\varepsilon) = N_V(\varepsilon)$. Specifically in the range $\text{diam}(L_n, d_{L_n}) < \varepsilon \leq \text{sep}(V, d_V)$, we need exactly one $\varepsilon$-ball for each $v \in V$ to cover $X_n$. Finally in the third case, we need $N_{L_n}(\varepsilon)$ $\varepsilon$-balls to cover $\{v\} \times L_n$ for each $v \in V$. This yields the stated estimate.

By the preceding claims, we now have the following for each $n \in \mathbb{N}, \varepsilon \in (0, D_n]$:

$$\rho(n, \varepsilon) \approx 4\varepsilon + \frac{1}{2n} \log_2(2N_{X_n}(\varepsilon)) = \begin{cases} 4\varepsilon + \frac{1}{2n} \log_2(2N_V(\varepsilon)) & : \varepsilon > \text{sep}(V), \\ 4\varepsilon + \frac{1}{2n} \log_2(2|V|) & : \text{diam}(L_n) < \varepsilon \leq \text{sep}(V), \\ 4\varepsilon + \frac{1}{2n} \log_2(2|V|N_{L_n}(\varepsilon)) & : \varepsilon \leq \text{diam}(L_n). \end{cases}$$

Notice that for sufficiently large $n$, $\inf_{\varepsilon > \text{diam}(L_n)} \rho(n, \varepsilon) = \rho(n, \frac{1}{n})$. Then we have:

$$\frac{1}{n} \leq E_n \leq B_n = \min_{\varepsilon \in (0, D_n]} \rho(n, \varepsilon) \leq \rho(n, \frac{1}{n}) \approx \frac{C}{n},$$

for some constant $C > 0$. Here the first inequality follows from the proof of Claim 2, the second from Theorem 8, and the third from our observation above. It follows that $E_n \asymp B_n \asymp \frac{1}{n} \to 0$. $\square$

**Remark 12.** Given the setup of the preceding proof, note that the Gromov-style bound behaves as:

$$G_n = \rho(n, 0) = \frac{1}{2n} \log_2(2|V|(n+1)) \approx C' \frac{\log_2(n+1)}{n},$$

for some constant $C' > 0$. So $G_n$ approaches 0 at a rate strictly slower than that of $E_n$ and $B_n$.

## 8 Discussion

We are motivated by a particular aspect of the numerical taxonomy problem, namely, the distortion incurred when passing from a metric to its optimal tree embedding. We describe and explore a duality between a tree embedding method proposed by Gromov and the well known SLHC method for embedding a metric space into an ultrametric tree. Motivated by this duality, we propose a novel metric space statistic that we call ultrametricity, and give a novel, tight bound on the distortion of the SLHC method depending on cardinality and ultrametricity. We improve this Gromov-style bound by replacing the dependence on cardinality by a dependence on doubling dimension, and produce a family of examples proving tightness of this dimension-based bound. By invoking duality again, we are able to improve Gromov's original bound on the distortion of his tree embedding method. More specifically, we replace the dependence on cardinality—a set-theoretic notion—by a dependence on doubling dimension, which is truly a metric notion.

Through Proposition 11, we are able to prove that our bound is not just asymptotically tight, but that it is strictly better than the corresponding Gromov-style bound. Indeed, Gromov's bound can perform arbitrarily worse than our dimension-based bound. We construct an explicit example to verify this claim in Appendix A, Remark 14, where we also provide a practical demonstration of our methods.

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
