[Supplementary Material]

## A   A Demonstration of Gromov's Embedding

We now present a practical demonstration of Gromov's embedding applied to a dataset. Figure 1 is a 720px by 960px `.bmp` image that visually looks like a tree. Each black pixel is interpreted as a point, and the collection of all 3798 points is endowed with the Euclidean distance. The resulting metric space is treelike with diameter 1 and hyperbolicity 0.0977. We take a dense subsample $X$ consisting of 500 points from this space, choose a point $p$ that maximizes $\|d_X - t_{X,p}\|_\infty$, and compute the Gromov embedding with respect to $p$. By our stability result (Theorem 6), the result of applying the Gromov embedding to the full dataset is faithfully represented by first taking such a subsample.

Because the data is treelike to begin with, we obtain $\|d_X - t_{X,p}\|_\infty = 0.1732$. For our upper bound, we use Inequality 2 from Theorem 7 and obtain 0.8686. In particular, the optimal $\varepsilon$ obtained in Inequality 2 was 0.0142, and the corresponding covering number was $N_X(\varepsilon) = 71$. Notice that without the stability and covering number bounds obtained in our work, one would be forced to use the bound in Gromov's embedding theorem (Theorem 3)—and due to the fact that there are 3798 points in the data, Gromov's bound of 1.2591 performs worse than even the trivial diameter bound.

| Quantity | Value |
|---|---|
| Diameter | 1 |
| Hyperbolicity | 0.0977 |
| $\|d_X - t_{X,p}\|_\infty$ | 0.1732 |
| Number of Points | 3798 |
| Gromov's Upper Bound | 1.2591 |
| Optimal $\epsilon$ | 0.0142 |
| $N_X(\epsilon)$ | 71 |
| Our Upper Bound | 0.8686 |

Figure 1: An image of a tree, with tabulated values of results obtained by applying Gromov's embedding, as described in §A. Notice that our upper bound performs significantly better than Gromov's bound in approximating the true additive distortion $\|d_X - t_{X,p}\|_\infty$. We remark that it has been claimed in [1] that Gromov's embedding is a 3-approximation to the optimal tree representation.

**Remark 13.** The supplementary material contains additional demos in Matlab format. The script `test_all.m` runs the computations described above on three images of trees. Two of these images depict trees with large hyperbolicity, and we have added them to illustrate situations where our bounds do not perform better than the trivial diameter bound.

**Remark 14.** It is easy to produce examples where the Gromov-style bound performs arbitrarily worse than our covering number upper bound. For an example, let $(V, d_V)$ be the space consisting of two points at distance 16, let $(L_n, d_{L_n})$ be the line metric space consisting of $n$ equally spaced points with diameter $\frac{1}{16}$, and let $X_n := V \times V \times V \times L_n$ be the space endowed with the following metric:

$$d_{X_n}\big((v_1, v_2, v_3, l), (v_1', v_2', v_3', l')\big) := \max\big(d_V(v_1, v_1'), \tfrac{1}{2}d_V(v_2, v_2'), \tfrac{1}{4}d_V(v_3, v_3'), d_{L_n}(l, l')\big),$$
$$\text{where } v_1, v_1', v_2, v_2', v_3, v_3' \in V, \ l, l' \in L_n.$$

Then by the proof of Claim 1 in Proposition 11, one can verify that $\mathrm{ult}(X_n, d_{X_n}) \approx \frac{1}{32}$, and that Gromov's bound is $G_n \approx \frac{1}{32}\log_2(16n)$, which grows without bound as $n \to \infty$. On the other hand, our covering number bound is never worse than $B_n \approx 4 + \frac{1}{32}\log_2(16)$ (using $\varepsilon = 1$ for $N_{X_n}(\varepsilon)$), which is a constant!

## B   Proofs

### B.1   Proof of Lemma 2

*Proof of Lemma 2.* Fix $\varepsilon \in (0, \mathrm{diam}(X)]$, and let $S$ be an $\varepsilon$-net in $X$. Note that $S \subset B(s, 2\,\mathrm{diam}(X))$ for any $s \in S$. By applying the doubling property once, this ball can be covered by $2^d$ balls of half the radius, and in particular, by applying the doubling property $k$ times, this ball can be covered by $2^{kd}$ balls of radius $\frac{2\,\mathrm{diam}(X)}{2^k}$. In particular, we may pick $k$ to be the smallest integer such that:

$$\frac{2\,\mathrm{diam}(X)}{2^k} \leq \frac{\varepsilon}{2}.$$

Then we must have:
$$\frac{2\,\mathrm{diam}(X)}{2^{k-1}} \geq \frac{\varepsilon}{2}.$$

And so $2^k \leq \frac{8\,\mathrm{diam}(X)}{\varepsilon}$. Thus the original ball containing $Y$ is now contained in $2^{kd}$ balls of radius $\leq \frac{\varepsilon}{2}$. Since any two points in $S$ are at least $\varepsilon$-separated by definition, it follows that each ball can contain only one point of $S$. Thus we have:
$$|S| \leq 2^{kd} \leq \left(\frac{8\,\mathrm{diam}(X)}{\varepsilon}\right)^d. \qquad \square$$

## B.2 Proof of Theorem 4

*Proof of Theorem 4.* First we prove the inequality, and subsequently we construct a sequence of metric spaces for which the error term asymptotically matches the Gromov-style bound.

*Proof of the inequality.* Let $\delta = \mathrm{ult}(X, d_X)$. First we claim that for any sequence of $2^k + 1$ points, we have:
$$\max_{1 \leq i \leq 2^k} d_X(x_i, x_{i+1}) \geq d_X(x_1, x_{2^k+1}) - k\delta.$$

To see this, we proceed by induction. Notice that for $k = 1$, the claim holds by the definition of $\mathrm{ult}(X, d_X)$. Let $k \in \mathbb{N}$, and suppose the claim holds for $k$. By the base case, we obtain:
$$d_X(x_1, x_{2^{k+1}+1}) \leq \max\left(d_X(x_1, x_{2^k+1}), d_X(x_{2^k+1}, x_{2^k+1+2^k})\right) + \delta$$

But by the induction step, we have:
$$d_X(x_1, x_{2^k+1}) \leq \max_{1 \leq i \leq 2^k} d_X(x_i, x_{i+1}) + k\delta$$
$$d_X(x_{2^k+1}, x_{2^k+1+2^k}) \leq \max_{2^k+1 \leq i \leq 2^{k+1}} d_X(x_i, x_{i+1}) + k\delta$$

Thus, taking the maximum of the two, we obtain:
$$d_X(x_1, x_{2^{k+1}+1}) \leq \max_{1 \leq i \leq 2^{k+1}} d_X(x_i, x_{i+1}) + (k+1)\delta.$$

This proves the claim. Next, let $x, x' \in X$. Let $c \in C_X(x, x')$. Write $c = \{x = x_1, \ldots, x_p = x'\}$. Note that if $c$ contains any repetition, i.e. if there exist $i < j \leq p$ with $x_i = x_j$, then we may replace $c$ by $c' = \{x_1, \ldots, x_i, x_{j+1}, \ldots, x_p\}$. Thus by reindexing if necessary, we obtain a chain of distinct elements $c' = \{x = x'_1, \ldots, x'_q = x'\}$, with $q < p$. Also note that $\mathrm{cost}_X(c') \leq \mathrm{cost}_X(c)$. Next let $k$ be the greatest integer such that $2^k \leq n$. Then we have $n \leq 2^{k+1} \leq 2n$. Since $c'$ has length $q \leq n$, we can define:
$$\bar{c} = \left\{x'_1, \ldots, x'_q, x'_q, \ldots, x'_q\right\},$$
where $\bar{c}$ is obtained from $c'$ by padding copies of the endpoint $x'_q$ until $\bar{c}$ has length $2^{k+1} + 1$. Notice that $\mathrm{cost}_X(\bar{c}) = \mathrm{cost}_X(c')$.

By applying the claim to $\bar{c}$, we obtain $\mathrm{cost}_X(c) \geq \mathrm{cost}_X(\bar{c}) \geq d_X(x, x') - (k+1)\delta$. Since $c$ was arbitrary, we also have:
$$\min_{c \in C(x, x')} \mathrm{cost}_X(c) = u_X(x, x') \geq d_X(x, x') - (k+1)\delta.$$

Since $x, x'$ were also arbitrary, we obtain:
$$\max_{x, x' \in X} \left(d_X(x, x') - u_X(x, x')\right) \leq (k+1)\delta \leq \log_2(2n) \cdot \mathrm{ult}(X, d_X). \qquad \blacksquare$$

*Proof of tightness.* We demonstrate tightness by constructing a sequence of finite metric spaces via a *snowflake* metric transform that realizes the logarithmic error rate. Recall that a *metric transform* is a continuous, monotone increasing, concave function $\Psi : \mathbb{R}_+ \to \mathbb{R}_+$ with $\Psi(0) = 0$; in particular, $\Psi$ maps metrics to metrics.

For any metric space $(X, d_X)$, we let $\Psi(X)$ denote $(X, \Psi(d_X))$. For spaces $X$ and transforms $\Psi(X)$ such that $\mathrm{ult}(\Psi(X)) \neq 0$, we define the following quantity:

$$R(\Psi) := \frac{\|\Psi(d_X) - \Psi(u_X)\|_\infty}{\mathrm{ult}(\Psi(X))}.$$

For any $x, x' \in X$, we also define:

$$d_X^{(1)}(x, x') := \min\left\{\max\left(d_X(x, z), d_X(z, x')\right) : z \in X\right\}.$$

This leads to the following reformulation of $\mathrm{ult}(X)$:

**Claim 4.**

$$\mathrm{ult}(X) = \|d_X - d_X^{(1)}\|_\infty.$$

*Proof.* If $\mathrm{ult}(X) = 0$, then it follows from the definitions that $d_X = d_X^{(1)}$. Suppose $\mathrm{ult}(X) > 0$.

$$
\begin{aligned}
0 < \mathrm{ult}(X) &= \max_{x,x',x''}\left\{d_X(x, x') - \max(d_X(x, x''), d_X(x'', x'))\right\} \\
&= \max_{x,x'}\max_{x''}\left\{d_X(x, x') - \max(d_X(x, x''), d_X(x'', x'))\right\} \\
&= \max_{x,x'}\left\{d_X(x, x') - \min_{x''}\max(d_X(x, x''), d_X(x'', x'))\right\} \\
&= \max_{x,x'}\left\{d_X(x, x') - d_X^{(1)}(x, x')\right\} \\
&= \|d_X - d_X^{(1)}\|_\infty.
\end{aligned}
$$

This shows the equality and proves the claim. ∎

Let $0 < \varepsilon \ll 1$, where we write $\ll$ to mean that $\varepsilon$ is much smaller than 1. Consider the *snowflake* metric transform $\Psi_\varepsilon(\alpha) = \alpha^\varepsilon$, see [9]. In the limit, when $\varepsilon \to 0$, all non-zero distances would become 1. That is, $\lim_{\varepsilon \downarrow 0} \Psi_\varepsilon(X)$ would be equal to the metric space with underlying set $X$ and the *discrete metric* (i.e. the metric attaining only the values 0 and 1). Note that the discrete metric is actually an ultrametric.

Next let $X$ be a finite set with $n > 1$ points, and let $E$ be a subset of $X \times X$ such that $G = (X, E)$ is a connected graph. Endow $G$ with edge weights 0 (for absence of an edge) or 1 (for presence of an edge). Let $(X, d_X)$ represent the finite metric space with $n$ points arising from computing the graph (or path length) distance on $G$. Specifically,

$$d_X(x, x') := \min\{|c| : c \in C_X(x, x')\},$$

where $C_X(x, x')$ is the set of all chains connecting $x$ and $x'$ over edges in the graph $G$. In this case, $u_X$, the SLHC output ultrametric, will be 1 between different points. Also note that $d_X$ takes integer values, and for any two points $x, x'$, we have $d_X^{(1)}(x, x') = \lceil \frac{d_X(x,x')}{2} \rceil$. Such a space will be called a *graph metric space*. In particular, we are interested in the *line graph metric spaces* $\overline{X_n}, n \in \mathbb{N}$. We define each $\overline{X_n}$ to be the graph metric space arising from the connected line graph having $(n+1)$ points $x_0, \ldots, x_n$ as vertices, and edges $(x_i, x_{i+1})$ for each $0 \leq i \leq n - 1$.

For $n \geq 2$, fix $\varepsilon_n = \frac{1}{\log^2(2n)}$ and consider the line graph metric space $\overline{X_{2n}}$ on $(2n + 1)$ points. Note that $\mathrm{diam}(\overline{X_{2n}}) = 2n$ for each $n$. Next let $X_{2n} = \Psi_{\varepsilon_n}(\overline{X_{2n}})$. Notice that the numerator of $R(\Psi_{\varepsilon_n})$ is now:

$$\max_{\alpha \in [0,2n]}\left(\alpha^{\varepsilon_n} - 1^{\varepsilon_n}\right) = (2n)^{\varepsilon_n} - 1.$$

By applying the reformulation of ultrametricity proved above, the denominator of $R(\Psi_{\varepsilon_n})$ becomes:

$$\max_{\alpha \in [0,2n]}\left(\alpha^{\varepsilon_n} - \left\lceil \frac{\alpha}{2} \right\rceil^{\varepsilon_n}\right) \approx \max_{\alpha \in [0,n]}\left(\alpha^{\varepsilon_n} - \left(\frac{\alpha}{2}\right)^{\varepsilon_n}\right) = (2n)^{\varepsilon_n}\left(1 - 2^{-\varepsilon_n}\right) = \left(\frac{2n}{2}\right)^{\varepsilon_n}\left(2^{\varepsilon_n} - 1\right).$$

Notice that equality holds above for even values of $n$. The expression for $R(\Psi_{\varepsilon_n})$ now becomes:

$$R(\Psi_{\varepsilon_n}) \approx \frac{((2n)^{\varepsilon_n} - 1)2^{\varepsilon_n}}{(2n)^{\varepsilon_n}(2^{\varepsilon_n} - 1)} = \frac{(2n)^{\varepsilon_n} - 1}{(2n)^{\varepsilon_n}} \cdot \frac{2^{\varepsilon_n}}{2^{\varepsilon_n} - 1} = \frac{e^{\varepsilon_n \log 2n} - 1}{e^{\varepsilon_n \log 2n}} \cdot \frac{2^{\varepsilon_n}}{e^{\varepsilon_n \log 2} - 1}$$

$$= \frac{e^{\frac{1}{\log 2n}} - 1}{e^{\frac{1}{\log 2n}}} \cdot \frac{2^{\varepsilon_n}}{e^{\frac{\log 2}{\log^2(2n)}} - 1}$$

Using a Taylor expansion, we see that for large $n$ this becomes $\approx \dfrac{\frac{1}{\log 2n}}{\frac{\log 2}{\log^2(2n)}} = \log_2(2n)$. $\blacksquare$

$\square$

## B.3 Proof of Theorem 5

*Proof of Theorem 5.* Let $x, x' \in X$ be such that:

$$|u_X(x, x') - u_X^A(x, x')| = |u_X(x, x') - u_A(\eta(x), \eta(x'))| = \|u_X - u_X^A\|_\infty.$$

Next let $c^* \in C_A(\eta(x), \eta(x'))$ be an optimal chain, i.e. a chain over points in $A$ such that $\mathrm{cost}_A(c^*) = u_A(\eta(x), \eta(x'))$. Write $c^* = \{\eta(x) = x_1, x_2, \ldots, x_n = \eta(x')\}$.

Next we define a new chain $c' \in C_X(x, x')$ by setting $c' = \{x = x_0, x_1, \ldots, x_n, x_{n+1} = x'\}$, i.e. $c'$ is just the composed chain $c' = x \circ c^* \circ x'$. Now we have:

$$u_X(x, x') \le \mathrm{cost}_X(c') = \max_{0 \le i \le n} d_X(x_i, x_{i+1})$$

$$\le \|d_X - d_X^A\|_\infty + \max_{0 \le i \le n} d_X^A(x_i, x_{i+1})$$

Note that $d_X^A(x_0, x_1) = d_X(\eta(x), \eta(x)) = 0$ and $d_X^A(x_n, x_{n+1}) = d_X(\eta(x'), \eta(x') = 0$, so:

$$= \|d_X - d_X^A\|_\infty + \max_{1 \le i \le n-1} d_X^A(x_i, x_{i+1}).$$

By our choice of $c^*$, each $x_i \in A$ for $1 \le i \le n$, so we have

$$= \|d_X - d_X^A\|_\infty + \max_{1 \le i \le n-1} d_A(x_i, x_{i+1})$$

$$= \|d_X - d_X^A\|_\infty + u_X^A(x, x').$$

Thus we obtain $u_X(x, x') - u_X^A(x, x') \le \|d_X - d_X^A\|_\infty$.

For the next part of the proof, let $c^* \in C_X(x, x')$ now be an optimal chain over points in $X$ beginning at $x$ and ending at $x'$. Write $c^* = \{x = x_1, \ldots, x_n = x'\}$. Then define $\eta(c^*) = \{\eta(x_1), \eta(x_2), \ldots, \eta(x_n)\}$. Note that $\eta(c^*) \in C_A(\eta(x), \eta(x'))$. Now we have:

$$u_X^A(x, x') = u_A(\eta(x), \eta(x')) \le \mathrm{cost}_A(\eta(c^*)) = \max_{1 \le i \le n-1} d_A(\eta(x_i), \eta(x_{i+1}))$$

$$\le \|d_X - d_X^A\|_\infty + \max_{1 \le i \le n-1} d_X(x_i, x_{i+1})$$

$$= \|d_X - d_X^A\|_\infty + u_X(x, x').$$

Thus we obtain $u_X^A(x, x') - u_X(x, x') \le \|d_X - d_X^A\|_\infty$. Finally we have $|u_X(x, x') - u_X^A(x, x')| \le \|d_X - d_X^A\|_\infty$. By our choice of $x, x' \in X$, it follows that:

$$\|u_X - u_X^A\|_\infty \le \|d_X - d_X^A\|_\infty. \qquad \square$$

## B.4 Proof of Theorem 6

*Proof of Theorem 6.* Let $x, x' \in X$ be such that:

$$|t_{X,p}(x, x') - t_{X,p}^A(x, x')| = |t_{X,p}(x, x') - t_{A,a}(\eta(x), \eta(x'))| = \|t_{X,p} - t_{X,p}^A\|_\infty.$$

Next let $c^* = \{x_1 = x, x_2, \ldots, x_n = x'\} \in C_X(x, x')$ be a chain over points in $X$ such that $g_{X,p}^{\mathcal{T}}(x, x') = \min_{1 \le i \le n-1} g_{X,p}(x_i, x_{i+1})$.

**Claim 5.** For any $x_i, x_{i+1} \in c^*$, $|g_{X,p}(x_i, x_{i+1}) - g_{A,a}(\eta(x_i), \eta(x_{i+1})| \leq \frac{3}{2}\|d_X - d_X^A\|_\infty$.

To verify Claim 5, note that:

$$2g_{X,p}(x_i, x_{i+1}) = d_X(x_i, p) + d_X(x_{i+1}, p) - d_X(x_i, x_{i+1})$$
$$\leq d_A(\eta(x_i), a) + d_A(\eta(x_{i+1}), a) + 2\|d_X - d_X^A\|_\infty - d_X(x_i, x_{i+1})$$
$$\leq d_A(\eta(x_i), a) + d_A(\eta(x_{i+1}), a) + 3\|d_X - d_X^A\|_\infty - d_A(\eta(x_i), \eta(x_{i+1}))$$
$$= 2g_{A,a}(\eta(x_i), \eta(x_{i+1})) + 3\|d_X - d_X^A\|_\infty.$$

Thus $g_{X,p}(x_i, x_{i+1}) - g_{A,a}(\eta(x_i), \eta(x_{i+1})) \leq \frac{3}{2}\|d_X - d_X^A\|_\infty$. To complete the proof of the claim, note that we can similarly obtain $g_{A,a}(\eta(x_i), \eta(x_{i+1})) - g_{X,p}(x_i, x_{i+1}) \leq \frac{3}{2}\|d_X - d_X^A\|_\infty$.

**Claim 6.** $|g_{X,p}^\mathcal{T}(x, x') - g_{A,a}^\mathcal{T}(\eta(x), \eta(x'))| \leq \frac{3}{2}\|d_X - d_X^A\|_\infty$.

To prove Claim 6, first note the following consequence of Claim 5:

$$g_{X,p}^\mathcal{T}(x, x') = \min_{1 \leq i \leq n-1} g_{X,p}(x_i, x_{i+1}) \leq \min_{1 \leq i \leq n-1} g_{A,a}(\eta(x_i), \eta(x_{i+1})) + \frac{3}{2}\|d_X - d_X^A\|_\infty$$
$$\leq g_{A,a}^\mathcal{T}(\eta(x), \eta(x')) + \frac{3}{2}\|d_X - d_X^A\|_\infty. \tag{6}$$

We now wish to prove an analogous inequality, but with the positions of $g_{X,p}^\mathcal{T}$ and $g_{A,a}^\mathcal{T}$ reversed. Let $c' = \{a_1 = \eta(x), a_2, \ldots, a_n = \eta(x')\} \in C_A(\eta(x), \eta(x'))$ be a chain such that $g_{A,a}^\mathcal{T}(\eta(x), \eta(x')) = \min_{1 \leq i \leq n-1} g_{A,a}(a_i, a_{i+1})$. Next define $c'' = \{x_1 = x, x_2 = a_2, x_3 = a_3, \ldots, x_n = x'\} \in C_X(x, x')$, and note that $\eta(c'') = c'$. Next we observe:

$$g_{A,a}^\mathcal{T}(\eta(x), \eta(x')) = \min_{1 \leq i \leq n-1} g_{A,a}(a_i, a_{i+1}) \leq \min_{1 \leq i \leq n-1} g_{X,p}(x_i, x_{i+1}) + \frac{3}{2}\|d_X - d_X^A\|_\infty$$
$$\leq g_{X,p}^\mathcal{T}(x, x') + \frac{3}{2}\|d_X - d_X^A\|_\infty. \tag{7}$$

Inequalities 6 and 7 together imply Claim 6. By Claim 6, we have:

$$|t_{X,p}(x, x') - t_{A,a}(\eta(x), \eta(x'))| = |d_X(x, p) + d_X(x', p) - 2g_{X,p}^\mathcal{T}(x, x')$$
$$- d_A(\eta(x), a) - d_A(\eta(x'), a) + 2g_{A,a}^\mathcal{T}(\eta(x), \eta(x'))|$$
$$\leq |d_X(x, p) - d_A(\eta(x), a)| + |d_X(x', p) - d_A(\eta(x'), a)|$$
$$+ 2|g_{X,p}^\mathcal{T}(x, x') - g_{A,a}^\mathcal{T}(\eta(x), \eta(x'))|$$
$$\leq \|d_X - d_X^A\|_\infty + \|d_X - d_X^A\|_\infty + 3\|d_X - d_X^A\|_\infty$$
$$= 5\|d_X - d_X^A\|_\infty.$$

By our choice of $x, x' \in X$, the result now follows. $\qquad \square$

## B.5  Proof of Claim 1 in Proposition 11

*Proof of Claim 1.* Recall that $V$ consists of two equidistant points, as in the endpoints of a unit interval. Let $n \in \mathbb{N}$, and let $x = (v, l), x' = (v', l'), x'' = (v'', l'') \in X_n$. Suppose first that not all of these points have the same $V$ coordinate. Without loss of generality, this means we can write $v \neq v' = v''$. Then we have:

$$\Psi_{X_n}^{\text{ult}}(x, x', x'') = d_V(v, v'') - d_V(v, v') = 1 - 1 = 0.$$

Here the first equality holds by the definition of $d_{X_n}$ and because $\text{sep}(V, d_V) > \text{diam}(L_n, d_{L_n})$, and the second equality follows because all points in $V$ are equidistant. Next, suppose that none of the $x, x', x''$ differ in the $V$ coordinate, i.e. $v = v' = v''$. Then we have:

$$\Psi_{X_n}^{\text{ult}}(x, x', x'') = d_{L_n}(l, l'') - \max\left(d_{L_n}(l, l'), d_{L_n}(l', l'')\right) \geq 0.$$

Maximizing over all choices of $x, x', x''$, it follows that $\text{ult}(X_n, d_{X_n}) = \text{ult}(L_n, d_{L_n})$. $\qquad \square$