[Reviews · NeurIPS 2016]

Reviewer 1

Summary

The paper is about embedding finite metric space in tree metric spaces. More specifically the paper studies the additive distortions of the so-called Gromov's embedding and the single linkage hierarchical clustering. The paper starts from the observation that the distortion given by Gromov depends on the cardinality of the embedded metric space, which is of course not satisfactory in the case of large point clouds. The paper provides new distortions bounds based on covering numbers and a notion of hyperbolicity (for the Gromov's embedding). The main idea behind these results is to consider as an intermediate construction some metrics induced by Voronoi partitions.

Qualitative Assessment

I have a mixed opinion about this paper. On one side, I find the subject very interesting. Moreover the paper is well written and pleasant to read. On the other side, I am not sure that the results of this paper are real advances for this field. Indeed the results are mostly based on known results and the stability results are not very surprising in my opinion. Moreover the authors are not able to discuss if there final bounds are tight are not. Applying the results to specific class of metric spaces would be interesting. Maybe such an important problem deserves additional work. Minor concerns : + Some definitions (2- or 3-approximations) are missing in the paper + Section 2 and some definitions given at the begining of Section 3 could be given in the Introduction

Confidence in this Review

2-Confident (read it all; understood it all reasonably well)


Reviewer 2

Summary

Embedding high-dimensional spaces into two dimensional trees is an important problem in data analysis. To measure the quality of such a tree, the authors suggest an approach which is based on the metric structure of the space rather than just the cardinality as has been done earlier. They provide an additive distortion bound for both general trees and ultrametric trees. The authors also show that by duality, the bounds apply to the single linkage hierarchical clustering.

Qualitative Assessment

The paper is written well, with Section 1 describing the problem and proposed approach effectively. The main contributions and claims also carry sufficient mathematical detail. However, I found Section 7 to be very short. It seems to end abruptly, without explaining the results. Moreover, further detail should be provide for remark 12, which says that in certain situations the additive bounds do not perform better than the trivial diameter bound. It would be useful to examine situations/disciplines where trees are expected to have large hyperbolicity.

Confidence in this Review

1-Less confident (might not have understood significant parts)


Reviewer 3

Summary

This paper proves distortion bounds for embedding metric spaces into tree metrics, in terms of hyperbolicity and doubling dimension, instead of just cardinality. The authors also show a duality between their results and single linkage hierarchical clustering.

Qualitative Assessment

Even though this bound is an improvement on Gromov’s upper bound, it is still quite loose. In the author’s demonstration (Figure 1), their bound appears to be barely non-trivial. Gromov’s upper bound is 1.23, the trivial bound is 1.0, this paper is 0.87 and the true distortion is almost an order of magnitude lower at 0.1732. This is the best result the authors show — in the other two examples (supplementary material) their bound is trivial. Though the author’s analysis is quite thorough, these results do not demonstrate the significance of their work. Perhaps an alternative demonstration or explanation would better motivate this paper.

Confidence in this Review

1-Less confident (might not have understood significant parts)


Reviewer 4

Summary

This paper provides new stability results and bounds on the distortion for the tree embedding of a metric space proposed by Gromov. It also exploits a duality between Gromov's embedding and the single linkage hierarchical clustering. As a consequence, stability and bounds are given for SLHC as well. The bounds are illustrated on simulations.

Qualitative Assessment

The paper is very interesting and provides good theoretical results. Such an analysis is welcome in the field. However, I think that it could be re-organized to improve its readibility and impact. There are repetitions which may hinder a smooth reading. (see examples below) In particular, I believe that the proofs could be put in the supplementary materials (although short sketch proofs can stay and give the main idea and steps). The definitions could be restricted only to what is absolutely needed. There are long math developments which may not be integrated enough into the global reasoning: the reader does not necessarily know why he needs to read this. This deletion/reorganization would give room for: (i) providing intuitive illustrations of metric spaces, trees, voronoi cells, doubling dimension; (ii) giving the intuition behind the theorems 5, 6, 7, 8 (intuition behind inegalities on D in Th.7,8?) and their importance; (iii) developing Section 7 to a page, (iv) providing more structure to the paper (subsections?), (v) write a short conclusion. For now, the intuition is given in "An overview of our approach" in the introduction, but not step by step in the core of the paper. Examples of repetitions: - Additive distortion is defined twice, in Introduction and Section 2. - The trivial bound is given in Introduction and Section 4. The introduction is maybe too technical? - Gromov's embedding is a 3-approximation to the optimal tree representation in Introduction and Section 7 Examples of unecessary math background: - many details on multiplicative distortion even though not used. Eg: what is the point of the first paragraph of Introduction and last paragraph of Sec. 2? Could they be shortened? - pages 4 and 5 could be embedded in a global reasoning, to keep the reader interested?

Confidence in this Review

1-Less confident (might not have understood significant parts)


Reviewer 5

Summary

The paper obtained an improved distortion bound of Gromov metric depending only on the hyperbolicity and doubling dimension of the metric. By leveraging the duality between Gromov’s embedding and SLHC embedding, the paper applied the conclusion of the distortion bound to SLHC that such additive bounds were previously unknown.

Qualitative Assessment

The paper has clear explanations for the problems, I have no questions to ask.

Confidence in this Review

1-Less confident (might not have understood significant parts)


Reviewer 6

Summary

While the most familiar form of embedding is Multi-dimensional Scaling, in which high-dimensional metric spaces are embedded into 2 or 3 dimensional Euclidean space, embedding into other spaces is also possible. This paper looks at embedding into a tree metric space, where the tree may or may not be ultrametric (having the same height from each leaf to the root), using Gromov's method. There is already a bound on the additive distortion between the original metric space and the embedding, but it depends on the cardinality of the data. The authors give a bound on the additive distortion that depends only on the hyperbolicity and doubling dimension of the metric space.

Qualitative Assessment

I find motivating this problem by phylogenetic trees construction (as in first sentence of abstract) to be a little misleading. While there are some distance based methods for constructing phylogenetic trees, these are primarily used as guide trees for sequence alignment or starting trees for maximum likelihood estimation or Bayesian methods based on evolutionary models. Thus getting the correct tree shape matters more than bounds on the distances between leaves. That being said, I think determining better additive bounds for Gromov's tree embedding method is a worthwhile goal, as there is a lot of data that is hyperbolic in nature, and thus embeds better into a tree than a low-dimensional Euclidean space. Indeed the authors mention this at line 66. The connection to Single Linkage Hierarchical Clustering is also interesting. I had trouble understanding the equation between lines 187 and 188, and how you are going from the clusters produced by SLHC to an ultrametric tree. What are the interior vertices of the tree? Are they always points from X? How does minimizing over the set of all chains in X fit into this? Minor comments: - l. 32: "tree" missing after "ultrametric" - l. 125: fix parentheses in fraction - l. 291: journal volume? - l. 300: capitalization and two "in"s

Confidence in this Review

1-Less confident (might not have understood significant parts)